# Transcriptional regulation and overexpression of GST cluster enhances pesticide resistance in the cotton bollworm, *Helicoverpa armigera* (Lepidoptera: Noctuidae)

Minghui Jin[1,4], Yan Peng[1,2,4], Jie Peng[1,2,4], Huihui Zhang[3], Yinxue Shan[1], Kaiyu Liu[3] & Yutao Xiao [1✉]

The rapid evolution of resistance in agricultural pest poses a serious threat to global food security. However, the mechanisms of resistance through metabolic regulation are largely unknown. Here, we found that a GST gene cluster was strongly selected in North China (NTC) population, and it was significantly genetically-linked to *lambda*-cyhalothrin resistance. Knockout of the GST cluster using CRISPR/Cas9 significantly increased the sensitivity of the knockout strain to *lambda*-cyhalothrin. Haplotype analysis revealed no non-synonymous mutations or structural variations in the GST cluster, whereas GST_119 and GST_121 were significantly overexpressed in the NTC population. Silencing of GST_119 or co-silencing of GST_119 and GST_121 with RNAi significantly increased larval sensitivity to *lambda*-cyhalothrin. We also identified additional GATAe transcription factor binding sites in the promoter of NTC_GST_119. Transient expression of GATAe in Hi5 cells activated NTC_GST_119 and Xinjiang (XJ)_GST_119 transcription, but the transcriptional activity of NTC_GST_119 was significantly higher than that of XJ_GST_119. These results demonstrate that variations in the regulatory region result in complex expression changes in the GST cluster, which enhances *lambda*-cyhalothrin resistance in field-populations. This study deepens our knowledge of the evolutionary mechanism of pest adaptation under environmental stress and provides potential targets for monitoring pest resistance and integrated management.

[1] Shenzhen Branch, Guangdong Laboratory of Lingnan Modern Agriculture, Key Laboratory of Gene Editing Technologies (Hainan), Ministry of Agriculture and Rural Affairs, Agricultural Genomics Institute at Shenzhen, Chinese Academy of Agricultural Sciences, Shenzhen, China. [2] College of Plant Science and Technology, Huazhong Agricultural University, Wuhan, China. [3] Institute of Entomology, School of Life Sciences, Central China Normal University, Wuhan, China. [4] These authors contributed equally: Minghui Jin, Yan Peng, Jie Peng. ✉email: xiaoyutao@caas.cn

Environmental adaptation evolution is the basic ability of insects to survive in diverse environments. Insecticide resistance in agricultural pests is a typical scientific issue of insect adaptation, posing immense challenges to global food security[1,2]. Agricultural pests can develop resistance to insecticides in many different ways, of which target site resistance and metabolic resistance through increased expression of detoxification enzymes are the main mechanisms[3,4]. Insect nicotinic acetylcholine receptors have been exploited as important molecular targets of spinosyns and neonicotinoids[5], acetycholinesterase as a target of organophosphorus and carbamate insecticides[6], and voltage-gated sodium channels as a target of organochlorine and pyrethroid insecticides[7]. Mutation of these targets often leads to a higher resistance in pests[6,8,9]. Metabolic resistance, which is more common than target site mutation, remains poorly characterized due to the complexity of this resistance mechanism[10]. The detoxification metabolism of pests is usually divided into three phases. In phase I, enzymes such as cytochrome P450 mono-oxygenase (P450s) and carboxylesterase act directly on the toxin molecule, catalyzing the oxidation, reduction, and chemical cleavage of toxins, thereby increasing the reactivity and hydrophilicity of the toxin[11,12]. In phase II, enzymes such as UDP-glycosyltransferases and glutathione S-transferases (GSTs) conjugate toxins with endogenous molecules, increasing the solubility of the toxins and decreasing the reactivity, resulting in the loss of the ability to diffuse across membranes[13,14]. In phase III, enzymes, such as ATP-binding cassette transporters, facilitate the active transport of toxins[15]. Variations in any of these stages may lead to insecticide resistance. The limited understanding of the mechanism of metabolic resistance has noteworthy implications for resistance monitoring and the management of agricultural pests.

The regulation of the expression of detoxification enzymes takes place at multiple levels, including promoter variation, transcription factor (TF) variation, and DNA methylation[16–19]. Variation in cis- or trans-regulatory elements could alter the expression of target genes, and thus induce resistance. In Drosophlia melanogaster, a TE insertion in the promoter sequence upregulated CYP6G1 expression[20]. In Anopheles funestus, cis-regulatory variants in the promoter region upregulated CYP6P9b expression, which is associated with pyrethroid resistance[4]. There is considerable evidence that has demonstrated that TFs are important players involved in the genetic mechanisms of resistance[16,21–25].

The cotton bollworm Helicoverpa armigera is one of the major agricultural pests worldwide, causing billions of dollars in losses every year[26,27]. The cotton bollworm is widely distributed in China, from the eastern coast to the northwestern interior. Due to geographical isolation, the cotton bollworm has formed two distinct geographical populations in North China (NTC) and Xinjiang (XJ) with few gene exchange[26]. The evolution of insecticide resistance in cotton bollworm is very rapid, and it is also the species with the most resistance events reported in Noctuidae (Arthropod Pesticide Resistance Database, Michigan State University), having evolved resistance to multiple insecticides, including carbamates, organophosphates and pyrethroids[28,29]. Lambda-cyhalothrin, a sodium channel modulator insecticide, has been widely used for pest control worldwide. In China, lambda-cyhalothrin is one of the most widely used insecticides, accounting for about a quarter of all applications[30]. Pyrethroid resistance has been attributed to target site mutations and/or increased insecticide detoxification. Target site mutation in the voltage-gated sodium channel gene has been associated with pyrethroid resistance in H. armigera[31]. The detoxification enzymes CYP450 and carboxylesterase have also been associated with metabolic resistance to pyrethroid in H. armigera[32].

Insecticide sensitivity monitoring of the North China and Xinjiang cotton bollworm populations revealed a high level of resistance to phoxim, cyhalothrin, and emamectin benzoate (EB) in the North China population[33]. However, the mechanisms underlying differences in insecticide sensitivity between populations have not been well understood.

In this study, we identified a GST cluster that was positively selected in the natural field-resistant strain from the NTC population by comparative analysis of population genomes. By constructing an $F_2$ genetic linkage population, we confirmed that the GST cluster was significantly associated with lambda-cyhalothrin (LC) resistance in the NTC population. We knocked out a 24 kb genomic fragment covering all six GST genes in the GST cluster using the CRISPR/Cas9 system, and the bioassay results showed that the sensitivity of the KO strain to LC was significantly increased. Then, we performed RNAi, promoter element analysis, and promoter activity determination to investigate the molecular basis of this GST cluster in insecticide response and found that variations between the NTC and XJ populations in the regulatory region of the GST genes resulted in complex expression changes in the GST cluster, which enhanced LC resistance in the NTC population.

## Results

**Differing sensitivity to insecticides between North China and Xinjiang populations of cotton bollworm**. To study the changes in sensitivity of cotton bollworm to insecticides in different regions, we downloaded and analyzed the insecticide monitoring results in NTC and XJ from the National Agro-Tech Extension and Service Center (2008–2019). The monitoring results showed that the field population of cotton bollworm in NTC had moderate or high resistance to phoxim (PX), high or extremely high resistance to LC, and susceptibility or moderate resistance to EB, while in Xinjiang it had susceptibility or low resistance to PX, susceptibility or moderate resistance to LC, and susceptibility or low resistance to EB (Fig. 1).

**Positive selection of the GST cluster in North China population**. In a previous study, we identified a region on chromosome 23 with strong selection signals using selective sweep analysis[33]. A GST cluster was located in this selective region. To further analyze the type of selection and to identify the potential causal mutation in the GST cluster locus, we scanned the local region of the GST cluster to examine the differences in nucleotide diversity and allele frequency divergence between the NTC and XJ populations. The $\theta_\pi$ (nucleotide diversity) results showed that the GST cluster locus in the NTC population had a lower level of nucleotide diversity than the XJ population, indicating that the GST cluster was selected in the NTC population (Fig. 2a). The NTC and XJ populations had similar $\theta_w$ values. The Tajma's D value of the GST cluster in the NTC population was less than zero, indicating that rare alleles were present at high frequency and confirmed that the GST cluster was positively selected in the NTC population (Fig. 2a). Gene scanning using LASSI revealed that selective peaks on the GST cluster were identified in the NTC population, whereas no selective peaks were identified in the XJ population, indicating that the selective sweeps of the GST cluster were the result of an independent evolutionary event (Fig. 2a). For the highest selection region, we found that the selection type in the NTC population was hard selection, indicating that the mutation of the GST cluster in the NTC population was a favorable mutation that was rapidly spread and fixed in the NTC population by positive selection (Fig. 2a). To confirm the detailed mutation type of the GST cluster, we analyzed the haplotypes of six GST genes in the GST cluster. The NTC and XJ populations

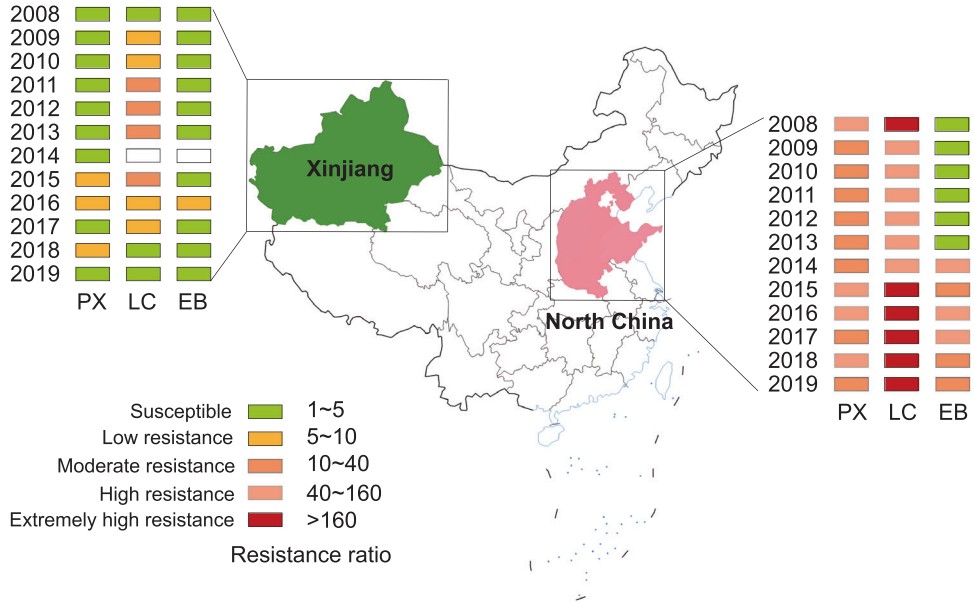

**Fig. 1 Insecticide monitoring results of cotton bollworm in North China and Xinjiang from 2008 to 2019.** *PX* phoxim, *LC* lambda-cyhalothrin, *EB* emamectin benzoate. Data source: National Agro-Tech Extension and Service Center (www.natesc.org.cn). White boxes represent no monitoring data.

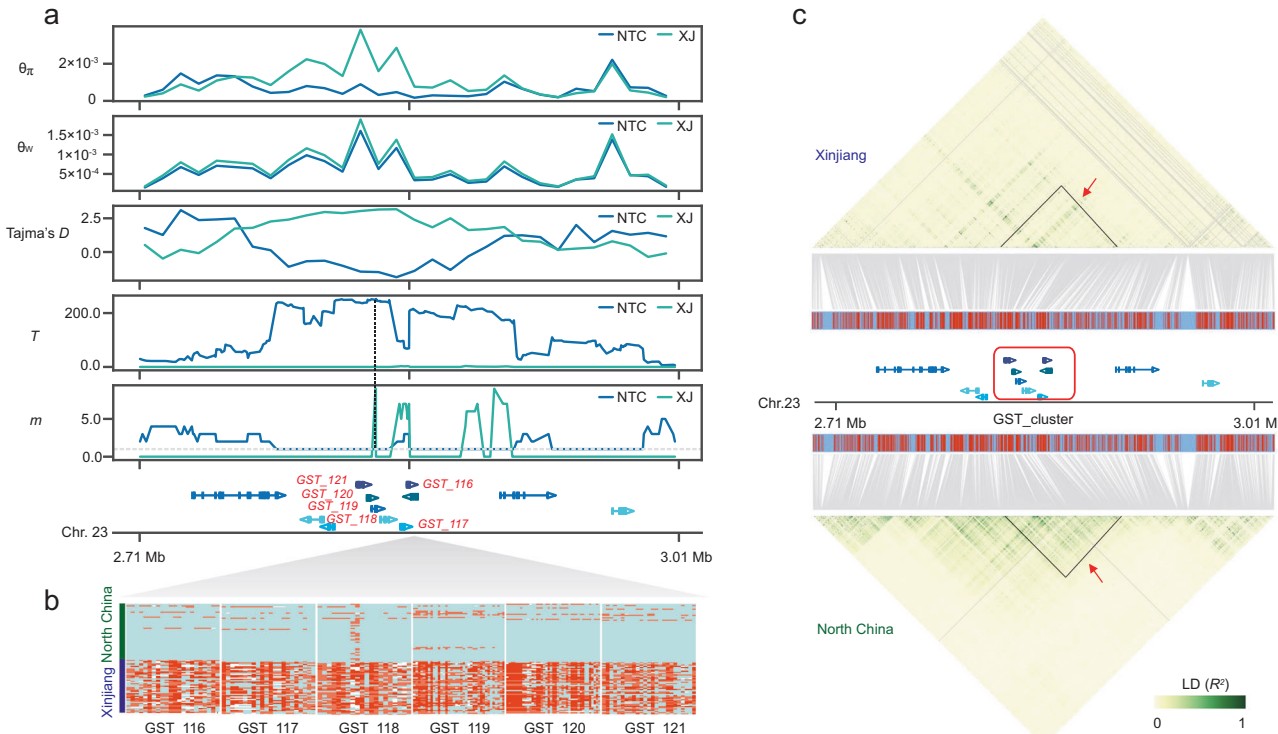

**Fig. 2 GST clusters under selective sweep in North China population. a** Selection signs on chromosome 23 (2.71–3.01 Mb), including the GST cluster. The putative sweep region was validated by $\theta_\pi$, $\theta_w$ and Tajima's D test. Positions of hard/soft sweep classification of the GST cluster were inferred by LASSI analyses. The peak point of the *T* value and its corresponding *m* value are marked by vertical dashed lines. Selective sweeps were classified as hard when $m = 1$ (marked by horizontal dashed lines) and soft when $m > 1$. Gene annotations in the sweep region are indicated at the bottom. **b** Haplotypes of six GST genes in the GST cluster. The genotype of the samples was categorized into two haplotypes (North China and Xinjiang). The presence of the wild allele and mutation allele is shown in blue and red, respectively. The white color symbolizes missing information. **c** Local LD heatmap of GST cluster region in Xinjiang, gene models, and local LD heatmap of GST cluster region in North China. The block of the GST cluster is marked with a red arrow.

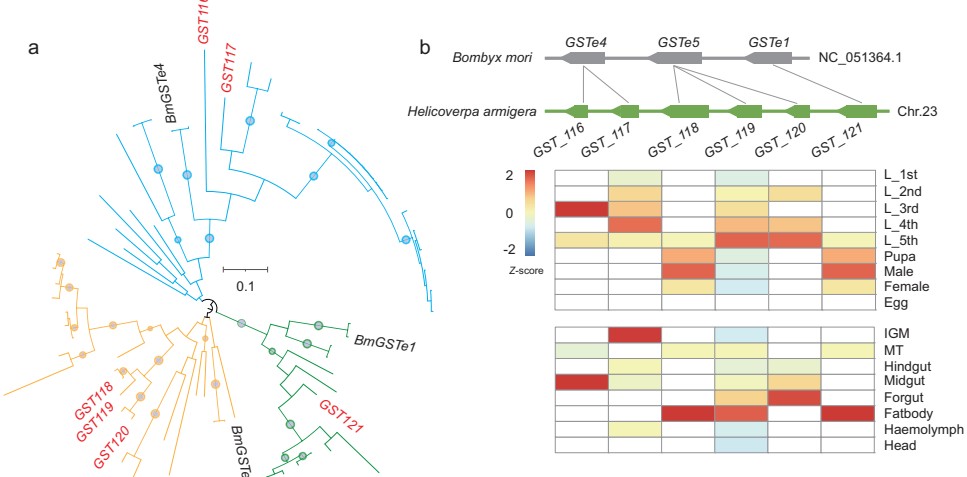

**Fig. 3 Expansion of the GST cluster. a** Maximum likelihood analysis of GST genes using 54 homologous genes. The homologous gene sequences were downloaded from NCBI, including *Bombyx mandarina*, *Bombyx mori*, *Manduca sexta*, *Arctia plantaginis*, *Trichoplusia ni*, *Spodoptera frugiperda*, *Spodoptera litura*, *Spodoptera littoralis*, *Chrysodeixis includens*, *Spodoptera exigua*, *Manduca sexta*, *Plodia interpunctella*, *Amyelosis transitella*, *Plutella xylostella* and *Helicoverpa zea*. Bootstrap support for representative nodes is shown. **b** Conserved syntenic genomic region containing the GST cluster in *H. armigera* and the ancestor species *B. mori*. Genes belonging to the same orthologous groups are connected by lines. Relative gene expression levels of GST genes in different developmental stages and tissues of *H. armigera* are shown using FPKM values.

showed two different haplotypes (Fig. 2b), but no non-synonymous mutations that could have caused amino acid variation were identified. In addition, linkage disequilibrium (LD) analysis showed that the LD block containing the GST cluster was more tightly linked in the NTC population (0.62) than in the Xinjiang population (0.15) (Fig. 2c).

**GST cluster formed by gene expansion.** Local gene duplication can contribute to the formation of gene clusters that are explicitly lineage-specific[34]. We analyzed the phylogenetic relationships of six GST genes in this gene cluster. The phylogenetic tree based on multiple homologous genes of these six GST genes showed that they were clustered in three clades (Fig. 3a), indicating a gene duplication event in this cluster. Synteny analysis between the ancestral species *Bombyx mori* (lepidopteran model organism) and the cotton bollworm showed that two expansions occurred by tandem duplication (Fig. 3b). The relative expression levels of these six genes showed diversified expression patterns at different developmental stages and in different tissues (Fig. 3b). Three-dimensional analysis revealed that all GST genes possessed the two canonical domains of the GST fold including the N- and C-terminus, and contained the GST_C_Delta_Epsilon domain (Supplementary Fig. 1).

**The GST cluster was genetically linked with *lambda*-cyhalothrin resistance.** To investigate whether the positively selected GST cluster in the NTC population was related to the previously mentioned high insecticide resistance trait in the NTC population, genetic linkage analysis was performed. First, an $F_2$ hybrid population was obtained by crossing a resistant NTC female with a susceptible XJ male (Fig. 4a). Ninety-six survivors (insecticide-resistant individuals) after diagnostic concentrations of PX, LC and EB treatment were selected and used for genetic linkage analysis. Genetic linkage analysis was performed using a molecular marker that distinguished the XJ and NTC populations (Fig. 4b). Comparing the proportion of each genotype ($rr$:$rs$:$ss$) among larvae fed on the control diet with that among resistant individuals fed on the insecticide-containing diet, we found that the proportions of genotypes fed the PX diet and EB diets fit the random assortment ratio 1:2:1 ($rr$:$rs$:$ss$ = 15:52:28, $\chi^2$ = 3.67,

df = 2, $P$ = 0.16; $rr$:$rs$:$ss$ = 20:56:20, $\chi^2$ = 2.67, df = 2, $P$ = 0.26) (Fig. 4c, d), indicating that the GST cluster was not linked to PX or EB resistance. In contrast, the GST cluster showed tight linkage with LC resistance ($rr$:$rs$:$ss$ = 60:24:12, $\chi^2$ = 72, df = 2, $P$ = 2.32 × 10$^{-16}$) (Fig. 4c, d).

**Knockout of the GST cluster using CRISPR/Cas9 significantly increased susceptibility to *lambda*-cyhalothrin.** To further verify that the GST cluster was associated with LC resistance, we generated a homozygous knockout strain using the CRISPR/Cas9 system. According to the genomic arrangement of the GST cluster, sgRNA1 and sgRNA2 were designed to target the genes at each end of the cluster, *GST_116* and *GST_121*, respectively (Fig. 5a). Primers of KO_F/R (for knockout screening), 117 F/R and 120 F/R (for homozygous screening) were designed for homozygous strain screening (Fig. 5a, b). After genotyping, six different knockout types were identified (Supplementary Fig. 2). Through a similar screening procedure[35], we obtained a 26 kb deletion homozygous knockout strain, named the GST-KO strain. Compared to its background strain (field-resistant strain collected from NTC, NTC-R strain), the sensitivity of the GST-KO strain to LC was significantly increased; however, there were no significant changes in its sensitivity to PX and EB (Fig. 5c). Although the sensitivity of the GST-KO strain to LC was significantly increased, the GST-KO strain still retained a low but significant resistance to LC compared to the susceptible 96 S strain (Fig. 5c).

**Promoter variation leads to differences in the expression of GST genes between North China and Xinjiang populations.** To further analyze the specific variation types in the GST cluster that led to insecticide resistance, we detected the expression level of genes in the GST cluster between the NTC and XJ populations, as no non-synonymous mutations were identified in the haplotype analysis. qPCR showed that the *GST_119* gene was more than sevenfold upregulated (*P* value < 0.01) and the *GST_121* gene was more than fourfold upregulated (*P* value < 0.01) in the NTC population compared to the XJ population (Fig. 6a). *GST_117* expression was significantly higher in the XJ population than in the NTC population. Other genes (*GST_116*, *GST_118*, *GST_120*) were low/not expressed or showed no significant change in

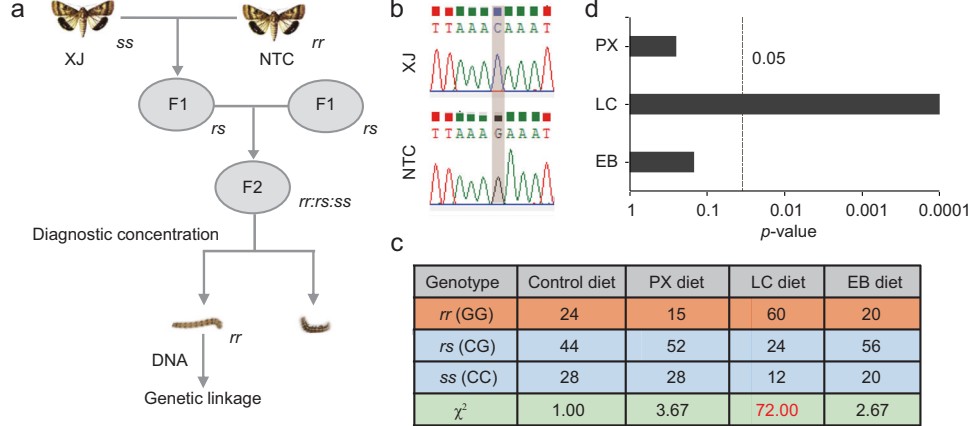

**Fig. 4 GST cluster genetic linkage with lambda-cyhalothrin resistance. a** Single-pair crosses between XJ and NTC virgin adults produced families of hybrid F$_2$ offspring. Progeny from F$_2$ families was exposed to diagnostic concentrations of phoxim (PX, 0.64 μg/cm$^2$), lambda-cyhalothrin (LC, 2.0 μg/cm$^2$), and emamectin benzoate (EB, 0.001 μg/cm$^2$). Ninety-six survivors of each treatment were selected for linkage analysis. **b** Molecular marker that distinguished between the XJ and NTC populations. **c** Number of survivors with different genotypes. $\chi^2$ test, based on the hypothesis of the random assortment ratio $rr$:$rs$:$ss$ = 1:2:1. $rr$ homozygote site in NTC, $ss$ homozygote site in XJ, $rs$ heterozygotes. **d** P values of three insecticides based on the $\chi^2$ test.

expression. The expression results were confirmed with RNA-seq (Supplementary Fig. 3). To determine whether the upregulation of *GST_119* and *GST_121* was sufficient to confer LC resistance, we knocked down their expression using RNAi (Supplementary Fig. 4). The bioassay showed that reducing *GST_121* expression did not significantly change the LC sensitivity of larvae, whereas reducing *GST_119* expression significantly increased the LC sensitivity of larvae (Fig. 6b). The injection of two equally mixed dsRNAs (dsGST_119 and dsGST_121) did not show a synergistic effect on the LC sensitivity of larvae (Fig. 6b). We also performed the in vitro functional verification of GST_119. The results showed that Sf9 cells transfected with the GST_119 plasmid had a significantly higher survival rate under LC treatment than Sf9 cells transfected with the blank plasmid ($p = 0.029$, df $= 4$, $t$ test), suggesting that GST_119 has the function of metabolizing LC in vitro (Supplementary Fig. 5). To interpret the interaction of GST_119 and LC, structural dynamics analysis was performed. The binding model was obtained by molecular docking (Supplementary Fig. 6). The carbonyl group and the cyano group of LC form three pairs of hydrogen bonds interacting with the hydroxyl groups of GLN92, ASN155, and GLN156 of GST_119, respectively.

To have a complete view of the potential regulatory elements driving the overexpression of *GST_119*, we amplified, cloned, and sequenced the promoter region between the resistant (NTC population) and susceptible (XJ population) populations. A number of SNPs and indels differentiated the upstream region of *GST_119* between the NTC and XJ populations (Supplementary Fig. 7). Predictive analysis of TF-binding sites revealed several differences between the NTC and XJ populations (Fig. 6c). The *CncC/Maf*, which has been reported to regulate the expression of detoxification enzymes, such as GST and CYP450, and to be associated with multi-insecticide resistance[17,36–40], was present in the NTC population but was absent in the XJ population. In addition, there were more *GATAe*-binding sites (15) in the NTC population than in the XJ population (10) (Fig. 6c). To demonstrate that the differences in the number of *GATAe* binding sites affected GST gene expression, we performed the luciferase assay in the *Sf9* cell line. GATAe promoted the transcription of PGL3-NTC_GST_119 and PGL3-XJ_GST_119, but there was a significant difference in promoting ability (Fig. 6d). The activity of the promoters in the NTC population was more than 6-fold higher than in the XJ population, which was

consistent with the fact that the NTC population had more GATAe binding sites. In addition, GATAe promoted the expression of PGL3-NTC_GST_121 and PGL3-XJ_GST_121. Although several SNPs and indels differentiated the upstream region of GST_121 (Supplementary Fig. 8), the luciferase assay showed no significant difference between the NTC and XJ populations, indicating that other mechanisms regulate its expression. These results suggest that the GATAe binding site is responsible for the transcriptional activity of the GST_119 promoter sequence and that the expansion of GATAe binding sites significantly increases the transcriptional activity of GST_119 in the resistant NTC population.

## Discussion

Population genomics is becoming a key tool for understanding the evolutionary mechanisms of insect adaptation[41,42]. The application of gene editing and other technologies supports further functional genomics research, and the combination of the two technologies will revolutionize theoretical innovation in this field.

Twelve years of field resistance monitoring data for cotton bollworm showed that the NTC population had a higher level of resistance to several insecticides than the XJ population. The difference in susceptibility to insecticides, particularly pyrethroids, between NTC and XJ may be due to several factors including crop structure and history of insecticide use. The NTC is a traditional agricultural farming region with a long history of crop cultivation, and the total amount and history of insecticide use are also relatively longer than those in the XJ region. Cotton bollworm in the NTC population has been under long-term selection pressure from insecticides, resulting in the rapid evolution of resistance[43]. In addition, XJ is a relatively closed region topographically, surrounded by mountains and deserts. Cotton bollworm in this region has limited gene flow with other populations. It is therefore difficult for resistance genes from outside the XJ population to infiltrate the XJ population.

Agricultural pests, especially polyphagous insects, usually have an efficient detoxification system to overcome xenobiotics, such as toxic plant defense chemicals[42]. This process usually involves several detoxification enzymes, including P450s, GSTs, CCEs, and UGTs. During species evolution, detoxification enzyme families have expanded in many pests to increase the ability of the species

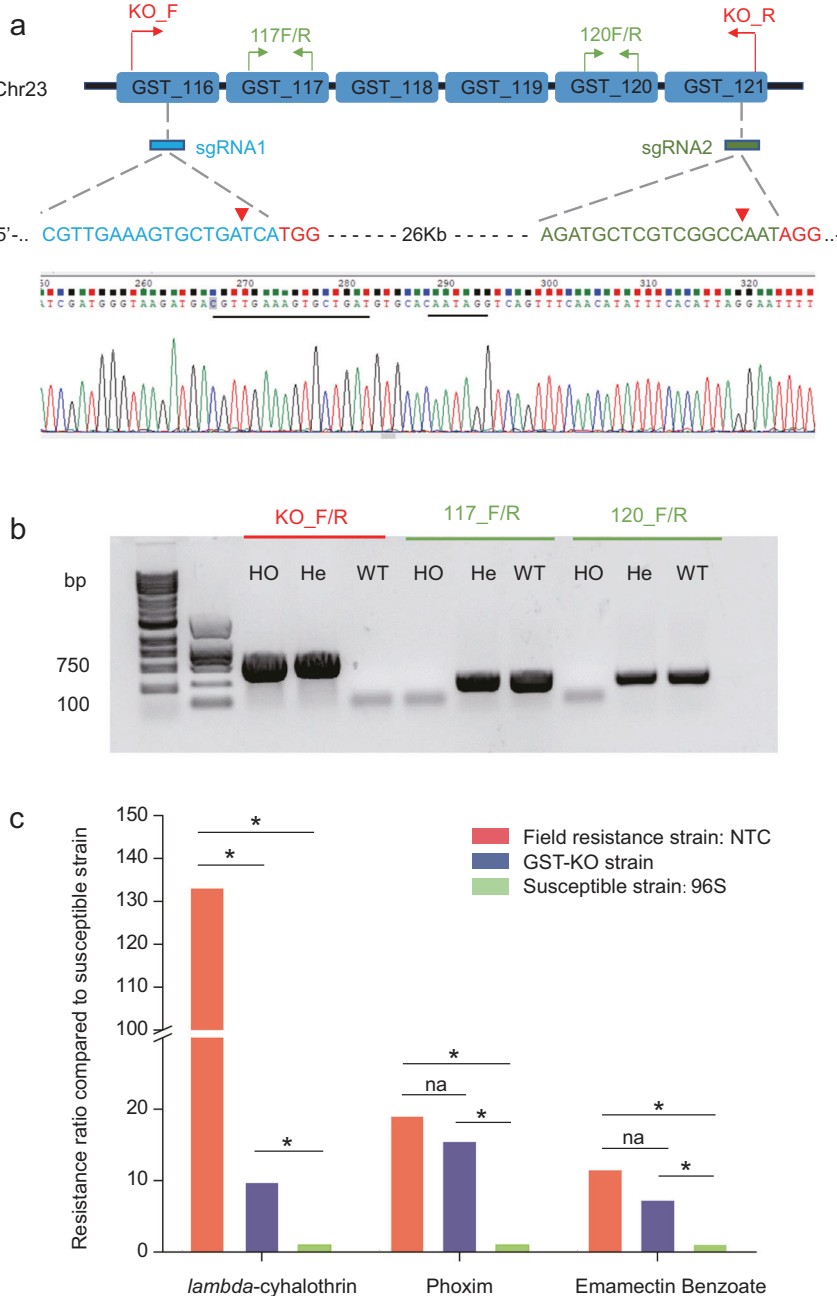

**Fig. 5 GST cluster knockout using the CRISPR/Cas9 system. a** Schematic diagram of sgRNA positions (sgRNA1 and sgRNA2), three pairs of detecting primers (KO_F/R, 117 F/R and 120 F/R), sgRNA target sequences and the PAM sequences (red) and chromatogram of the mutation type, a 26 kb deletion. The cutting sites of the Cas9 protein are indicated with red triangles. **b** Genotyping of individuals for deletion of the GST cluster according to the banding patterns of the PCR products amplified with three primer pairs. HO homozygote, He heterozygote, WT wild type. **c** Bioassay results of field-resistant NTC strain, GST-KO strain, and susceptible 96 S strain treated by three insecticides lambda-cyhalothrin, phoxim, and emamectin benzoate. The GST-KO strain was derived from the NTC strain by knocking out the GST cluster. Resistance ratios were calculated as the $LC_{50}$ of NTC/$LC_{50}$ of 96 S and the $LC_{50}$ of GST-KO/$LC_{50}$ of 96 S. Asterisks indicate the resistance ratios that were significantly different between the two strains based on their fiducial limits of $LC_{50}$ values, which did not overlap. na, no significant change.

to metabolize xenobiotics[10,44,45]. The expansion of the metabolic enzyme family not only increases the host range of pests, but also provides efficient and diverse integrated ways to cope with pesticide stress, significantly increasing the speed and intensity of resistance evolution. Different genes of the cotton bollworm CYP6AE gene cluster have the ability to metabolize the same xenobiotics, *CYP6AE11*, *CYP6AE14* and *CYP6AE19*, which have the ability to metabolize 2-tridcanone[46]. GST metabolic enzymes

are important detoxification enzymes in insects, and they play an important role in the metabolism of xenobiotics[17,47–49]. The GST gene cluster (six GST genes) that was positively selected in the NTC population, evolved from the expansion of three GST genes, and six GST genes were expressed in different developmental stages or tissues, confirming that all six GST genes were not pseudogenes. In addition, protein structure prediction confirmed that they all had the conserved GST domain. The amino acid

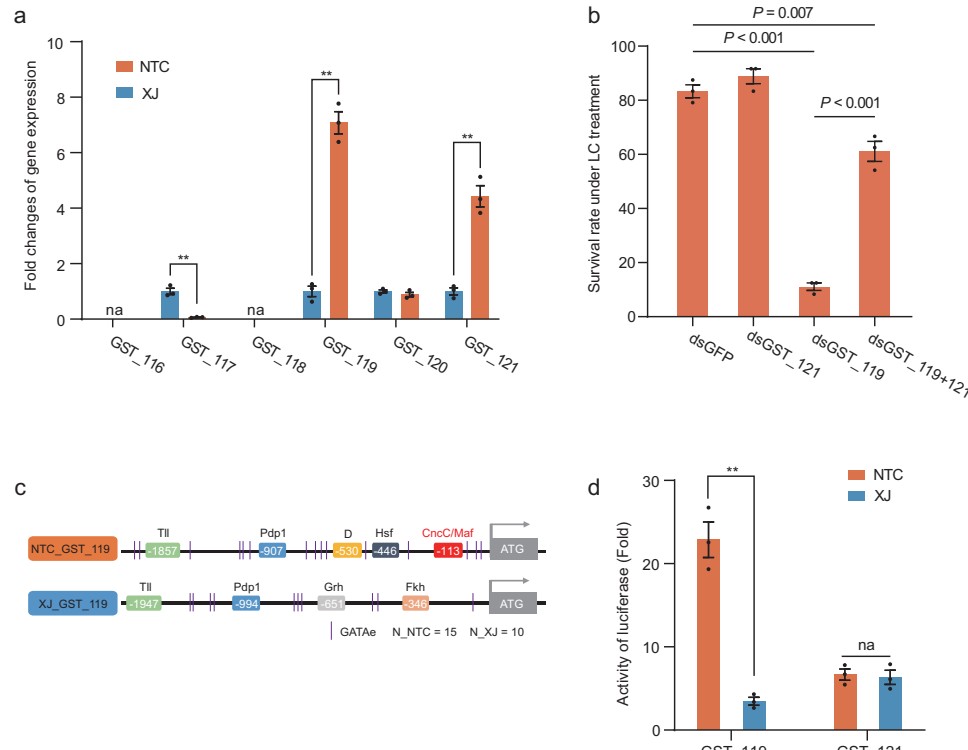

**Fig. 6 Mutations in the promoter of GST_119 enhance the expression of this GST gene, resulting in resistance in the NTC population. a** Relative expression of each GST gene in the resistant (NTC, $n = 3$) and susceptible strain (XJ, $n = 3$) of cotton bollworm, as determined by RT-qPCR. Error bars represent standard deviation. A significant difference in expression between the susceptible and resistant strains is indicated using an asterisk (Student's $t$ test, **, $P < 0.01$). na means that no expression was detected. **b** Effects of injecting dsRNA on the mortality of larvae treated with lambda-cyhalothrin ($n = 3$). dsGFP ($n = 3$) was used as a control. Each third-instar larva was injected with 1.5 μg dsRNA. Twenty-four larvae were used as a treatment, with three replicates. The concentration of lambda-cyhalothrin was 100 μg/mL. Larval mortality rates were recorded 3 days after treatment. The P value on the error bar indicates a significant difference analyzed by Student's $t$ test. Error bars represent standard deviation. **c** Prediction of transcription factor binding sites in the promoter region of GST_119 from North China population (NTC) and Xinjiang population (XJ). The nucleotides are numbered relative to the translation start site (ATG), with the sequence upstream of it preceded by "-". The transcription factor binding sites are shown in different colors. GATAe binding sites are represented by vertical lines. (**d**) Dual-luciferase assay. The pGL3 GST_119 and GST_121 promoter constructs of the NTC or XJ population were transfected with GATAe constructs into sf9 cells. The cells were collected 48 h after transfection, and luciferase activities were measured ($n = 3$). Error bars represent standard deviation. A significant difference in the activity of luciferase between the NTC and XJ populations is indicated using asterisks (Student's $t$ test, **, $P < 0.01$). na, no significant changes.

sequences of these GST genes showed high homology, and this gene cluster may contribute to the superposition of the metabolic capacity of certain chemicals in the cotton bollworm.

Constitutive quantitative changes in the expression of detoxification enzymes are common mechanisms underlying insect resistance to xenobiotics[50]. The expression of GST_119 and GST_121 genes was significantly upregulated in the NTC population. We speculated that cotton bollworm may comprehensively upregulate the GST_119 and GST_121 genes in the GST cluster to achieve high LC resistance. Upregulation of the GST_119 gene may play an important role in LC resistance, as GST_119 expression varied more between NTC and XJ populations, and the absolute expression of GST_119 was 150-fold higher than that of GST_121. After knockout of the GST cluster, the LC sensitivity of the larvae changed significantly. Similarly, knockdown of GST_119 expression by RNAi also significantly altered larval LC sensitivity, but the effect was much smaller than that of cluster knockout. We speculated that part of the reason was the limited efficiency of RNAi, which only partially downregulated the target gene. The main reason may be that other genes in the cluster have synergistic effects with GST_119, enhancing the metabolic capacity of LC. Similar synergistic effects between homologous genes have been reported in studies of Bt resistance in cotton bollworm[51,52]. To verify whether there was a synergistic effect

between GST_119 and GST_121, we simultaneously injected dsGST_119 and dsGST_121 to knockdown the expression of both genes. However, the interference efficiency of Lepidoptera is limited, and we could only inject them during the small larval stage to ensure interference efficiency, so the amount of dsRNA injected was limited. When dsGST_119 and dsGST_121 were injected together, the interference efficiency of both genes was low. Better methods, such as double knockout of GST_119 and GST_121 combined with in vitro metabolic verification, are needed to investigate this hypothesis in the future. In addition, the expression level of GST_117 was significantly decreased in the NTC population, which may be a feedback of functional redundancy caused by the high expression of GST_119 and GST_121. In conclusion, we speculated that the adaptation of cotton bollworm to high-intensity insecticide spraying environments in NTC was related to the comprehensive regulation of GST gene clusters.

Insecticide resistance caused by the variation in metabolic enzymes has been more widely reported than that caused by target mutation[3,6,50]. However, most studies have only confirmed the association between resistance and the expression level of the corresponding detoxification gene, and few studies have focused on the mechanism of this transcriptional change. In this study, the GST_119 promoter sequence of the NTC population was

significantly different from that of the XJ population, with the NTC population having more GATAe transcription factor binding sites. The dual-luciferase reporter assay confirmed that the GATAe transcription factor regulates the expression of GST_119 and GST_121. Furthermore, in the presence of the GATAe transcription factor, the promoter activity of GST_119 in the NTC population was six times higher in NTC population than that in the XJ population. It was confirmed that promoter variation in the NTC population was an important factor leading to high GST_119 expression. In addition, we found that the promoter of GST_119 from the NTC population had the *CncC/Maf* transcription factor binding site, whereas the promoter sequence of GST_119 from the XJ population did not have this transcription factor binding site. The *CncC/Maf* transcription factor has been reported to regulate the expression of several metabolic enzymes, including P450s and GSTs[4,16,53,54], and thus to be involved in the generation of insecticide resistance[21,53,54]. In addition to *CncC/Maf*, the TF *AhR/ARNT* has also been reported to regulate the expression of P450 and GSTs[55,56]. Further research is required to determine whether this TF is also involved in the differential expression of the GST cluster. Therefore, the presence or absence of this transcription factor may also affect the expression of GST_119.

Genetic linkage experiments, it was confirmed that the GST gene cluster in NTC population was closely linked to LC resistance, but the linkage was not 100%. In addition, after knocking out of the GST gene cluster in the NTC population, KO individuals were significantly more sensitive to LC but still retained about 10-fold higher LC resistance than sensitive strains. These results suggest that a complex resistance mechanism exists in the NTC population. In addition to the GST gene cluster, which is a major locus, other loci are still involved in insecticide resistance in the NTC population. As a type of broad-spectrum metabolic enzyme, GSTs may have metabolic or chelating effects on other types of pesticides or plant secondary metabolites other than LC. This pathway variation may provide further unexplored enhancements in the environmental adaptability of the cotton bollworm.

In conclusion, we have identified a GST gene cluster that underwent positive selection in NTC. Genetic linkage analysis has demonstrated that this GST gene cluster is associated with resistance to LC. Knockout of the GST gene cluster by CRISPR/Cas9 confirmed that this GST gene cluster is involved in LC metabolism. The complex regulatory mechanism of the GST gene cluster leading to insecticide resistance in the NTC population was then elucidated by RNAi, promoter sequence analysis, and a dual-luciferase assay (Fig. 7). These findings can enhance comprehension of insecticide resistance mechanisms in natural populations and furnish potential targets for the monitoring and management of pest resistance.

## Methods

**Insect strains.** The *H. armigera* laboratory strain 96 S was started from 20 pairs of adults, collected from a conventional cotton field located in Xinxiang, Henan Province, China, in 1996[57]. The 96 S strain was reared under laboratory conditions using an artificial diet without the addition of any insecticides. Two field-collection strains, NTC (30 pairs) and XJ (30 pairs), were collected from Xiajin, Shandong Province (North China) and Korla, Xinjiang Uygur Autonomous Region, respectively, in 2021. All strains were reared at 26 ± 1 °C with a 60 ± 10% relative humidity and 14:10 h light: dark. For adults, a 10% sugar solution was supplied to replenish energy.

**Source of field resistance monitoring results.** The insecticide resistance monitoring data of *H. armigera* were downloaded from the National Agro-Tech Extension and Service Center (www.natesc.org.cn), and the NATESC staff provided some year's data as a supplement. The data on planting structure, insecticide usage history was obtained from the National Bureau of Statistics (www.stats.gov.cn).

**Detection of selective sweeps.** We used 1,051,382 high-quality SNPs from our previous study[33] for selective sweep analysis. The brief flow of SNP calling was: quality control of raw reads, aligned against the reference genome, perform SNP calling using GATK with the parameter: QD ≥ 2.0 && MQ ≥ 40.0 && FS ≤ 60.0 && SOR < 3.0 && MQRankSum ≥ −12.5 && ReadPosRankSum ≥ −8.0. VCFtools was used to retain only biallelic SNPs showing no >10% missing data. Plink was used to subset SNPs with minor allele frequency ≥0.05. Then, the final subset of 1,051,382 SNPs was annotated using snpEff. The sequencing data is available at NCBI under BioProjects SAMN18253696 and PRJNA713413.

In the population differentiation method, $\theta_\pi$, $\theta_w$, and Tajma's D were calculated in 10-Kb non-overlapping windows in the 300 Kb region (located at 2.71–3.01 Mb in Chromosome 23) containing the GST cluster using VariScan (v2.0.3)[58]. In the hard and soft sweep analysis method, we used program LASSI[59] to calculate $T$ and $m$ with a window size and step size of 50 and 5 SNPs, respectively. Selective sweeps were identified through the $T$-statistic, as defined by the log CLR of the sweeps model in comparison to the genome-wide background-a neutrality model. Regions where the $T$ value reaches its peak are considered to be the most likely locations for the occurrence of beneficial variation. For regions exhibiting elevated $T$ values, it was determined that sweeps were classified as either hard or soft based on the model parameter $m$, which denotes the number of sweeping haplotypes ($m = 1$, hard; $m > 1$, soft). LD was analyzed using the program Plink (v1.90)[60].

**Phylogenetic analysis.** A total of 54 amino acid sequences of GSTs of *Bombyx mori*, *Bombyx mandarina*, *Manduca sexta*, *Trichoplusia ni*, *Spodoptera frugiperda*, *Spodoptera litura*, *Spodoptera littoralis*, *Plutella xylostella*, *Ostrinia furnacalis* and *Helicoverpa zea* were downloaded from NCBI. All amino acid sequences of GSTs were aligned using Clustal and converted to a MEG file using MEGA 7.0.26. The phylogenetic tree was then constructed using the maximum likelihood method in MEGA with a bootstrap of 1000 replicates.

**Developmental stage- and tissue-specific expression of GST cluster.** To determine the developmental stage and tissue-specific expression of six GST genes in the GST cluster, we re-analyzed our previously published RNA-seq data from cotton bollworm[61]. After filtering by fastp, the clean reads were mapped to our reference genome using HISTA2.2.4 with "-ran-standness RF" and other parameters set as a default. The FPKM values of nine developmental stages and eight tissues were calculated to quantify the expression abundance using RSEM software and used for the heatmap by R software.

**Structural modeling.** The 3D model of GST was constructed using Phyre 2.0 (www.sbg.bio.ic.ac.uk/phyre2/html) and visualized on the web server EzMol (http://www.sbg.bio.ic.ac.uk/ezmol/). The confidence and coverage of each GST structure were > 99.7% and > 95%, respectively.

**Population construction for genetic linkage.** To generate an $F_2$ segregating family to analyze whether the GST cluster was genetically linked with insecticide resistance, an NTC male (field population collected from North China, resistance) was crossed

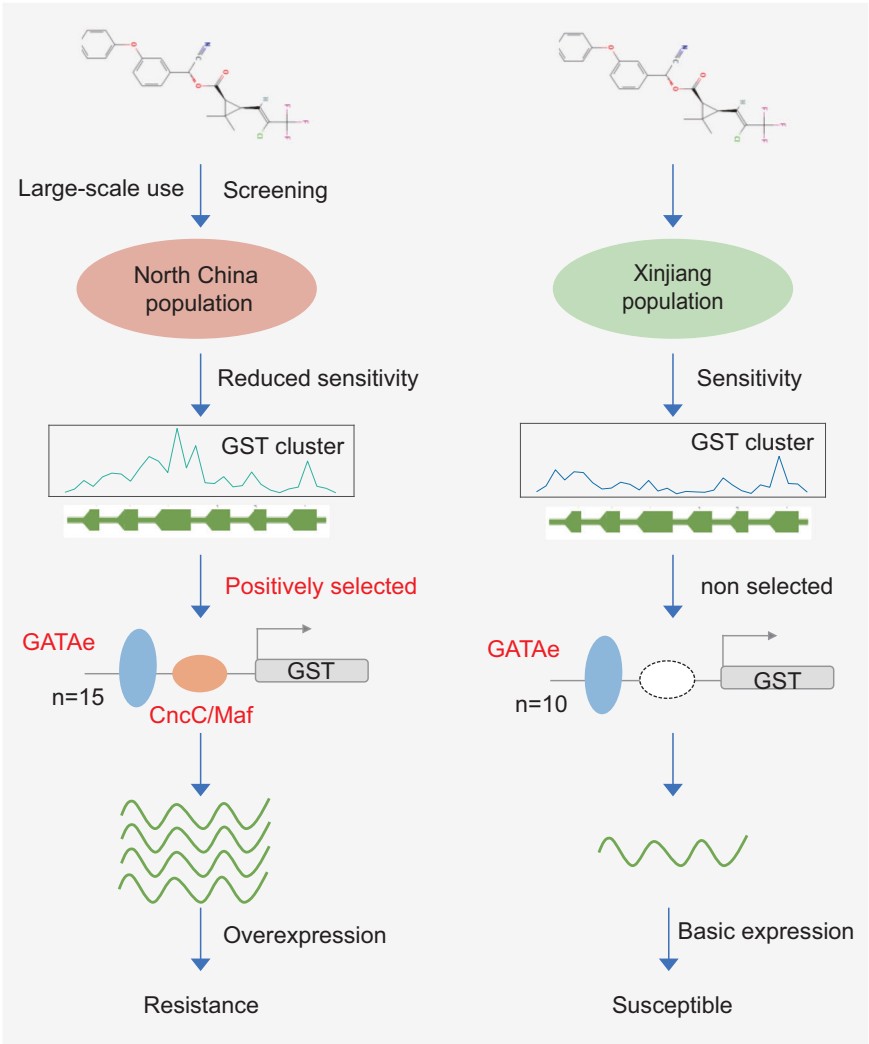

**Fig. 7 Schematic of the promoter variation leading to overexpression of GST genes, resulting in insecticide resistance in a natural field population.** In the naturally resistant population from North China, the GST cluster was positively selected. Mutation of the promoter enhances the expression of the GST gene, leading to insecticide resistance.

with an XJ female (field population collected from XJ, susceptible) to generate an F₁ progeny. Then F₁ progeny were reared to pupa. After eclosion of F₁ moths, 100 pairs of male and female moths were intercrossed to produce F₂ segregating generation.

**Phenotypic screening**. The third-instar F₂ larvae were collected for phenotypic screening. Based on the bioassay results, the diagnostic concentrations of PX ($0.64\,\mu g/cm^2$), LC ($2\,\mu g/cm^2$), and EB ($1\,ng/cm^2$) were selected. Then, the diagnostic concentrations of different insecticides were added to the artificial diet and solvent-only diet as a control. Ninety-six individuals were used for each treatment. The number of deaths and survivors was recorded after 3 days, and samples were retained for DNA extraction.

**Linkage ratio calculation**. For the linkage ratio calculation, we first identified a marker of the GST cluster (CC in XJ, GG in NTC). After extracting the survivors' DNA, we detected the genotype of each individual. We used Chi-square tests for the genetic linkage of the GST cluster to insecticide resistance based on the hypothesis of the random assortment ratio $rr:rs:ss = 1:2:1$.

**sgRNAs design and synthesis**. To further analyze the function of the GST cluster, we knocked out the GST cluster using the CRISPR/Cas9 system. A pair of sgRNAs were designed using the sgRNAcas9 design tool (Supplementary Table 1)[62]. sgRNA1 and sgRNA2 were located on GST_116 and GST_121, respectively. The sgRNA target sequences were checked in a search of the *H. armigera* genome using the sgRNAcas9 design tool, and no potential off-target sites were identified. A PCR-based approach was used to prepare sgRNA according to the manufacturer's instructions (GeneArt Precision gRNA Synthesis Kit, Thermo Fisher Scientific, USA). The template DNA for the in vitro transcription of the sgRNAs was made using PCR-based fusion of two oligonucleotides with the T7 promoter (Target F: TAATACGACTCACTATAG + the target sequence; Target R: TTCTAGCTCTAAAAC + the reverse complementary sequence of the target). The PCR conditions were as reported by Jin et al.[35]. The sgRNAs were synthesized by in vitro transcription using the GeneArt Precision gRNA Synthesis Kit, according to the manufacturer's instructions. The Cas9 protein (GeneArt Platinum Cas9 Nuclease) was purchased from Thermo Fisher Scientific (Shanghai, China).

**Embryo collection and injection**. Freshly laid eggs of the North China population (within 2 h after oviposition) were washed from

gauze using 1% (v/v) sodium hypochlorite solution and rinsed with distilled water. The eggs were then stuck on a microscope slide with double-sided adhesive tape. About 1.5 nL of a mixture of two sgRNAs (150 ng/μL) and Cas9 protein (50 ng/μL) was injected into each embryo using Nanoject III (Drummond, Broomall, USA). The injected eggs were incubated at 25 °C and 65% RH for hatching.

**Mutagenesis detection.** According to the GST_116 and GST_121 gene orientations on chromosome 23 and the sgRNA locations within each gene, the forward-detecting primer KO_F (located in GST_116) and the reverse primer KO_R (located in GST_121) were designed to detect GST cluster deletion (Fig. 5a, Supplementary Table 1). A small fragment of genomic DNA (~550 bp) was expected to be amplified with the primer pair KO_F/KO_R if the GST cluster was deleted. To determine whether the GST cluster deletion was homozygous or heterozygous, two pairs of primers were designed in GST_117 (117 F/R) and GST_120 (120 F/R) with the GST cluster. The genotype of the GST cluster deletion was identified according to the banding pattern of the PCR-amplified products (Fig. 5b). Three steps were taken to obtain hereditary homozygote: (i) reciprocal crossed between moths of $G_0$ and wild type moth in single pair, and the mutation type of $G_0$ moths was determined after laying enough eggs; (ii) $G_1$ larvae of the selected mutation type were reared to pupa, and genotype of larvae was determined using their exuviates; individuals carried the mutated allele were mass-crossed to produce $G_2$; (iii) $G_2$ individuals were genotyped, and homozygotes were selected to produce a knockout strain.

**Insecticide bioassay.** The PX (96% active ingredient), LC (95% active ingredient), and EB (95% active ingredient) used in this study were provided by Shandong Agricultural University, China. We used diet overlay bioassays to test third-instar larvae against these three insecticides. Each insecticide was dissolved in DMSO, and then, gradient insecticide concentrations were prepared. An artificial diet (900 μL) was dispensed into 24-well plates (surface area = 2 cm²). After the diet solidified, 50 μL of the insecticide solution was applied to the surface of the diet in each well. DMSO was used as the control. Mortality was recorded after 3 days. The half-lethal concentration ($LC_{50}$) that killed 50% of the tested larvae and the corresponding 95% confidence interval were calculated using Probit analysis of the mortality data in SPSS. Two $LC_{50}$ values were considered significantly different if their 95% confidence interval did not overlap.

**qRT-PCR.** Total RNA from the fat body tissue of 4th instar larvae was extracted using Trizol reagent (Invitrogen, USA). The cDNA was synthesized using TranScript One-Step gDNA Removal and cDNA Synthesis SuperMix (TransGen Biotech, China). Gene-specific primers were designed using Beacon Designer software (Supplementary Table 1). qRT-PCR was performed on a StepOnePlus Real-Time PCR System. The relative gene expression level was calculated according to the $2^{-\Delta\Delta Ct}$ method[63] using *Actin* and *GAPDH* genes as internal controls[64]. Three replicates were performed for each gene.

**RNAi.** Double-stranded RNA (dsRNA) of GST_119 and GST_121 (Supplementary Table 1) was prepared in vitro using a T7 RNAi Transcription Kit (Vazyme Biotech, China) following the manufacturer's instructions. The quality of the dsRNA was assessed using 1.5% agarose gel electrophoresis, and the concentrations were measured using a Nanodrop 2000c spectrophotometer. Approximately 2.5 μg of dsRNA was injected into the third-instar larvae of *H. armigera*. dsGFP was injected as a control. The injected larvae were placed individually into a 24-well plate, provided with an artificial diet for 24 h to recover, and then provided with an artificial diet with LC (0.5 μg/cm²). Each treatment consisted of 24 larvae with 3 replicates ($n = 24 \times 3$). Transcription of GST_119 and GST_121 was detected 48 h after dsRNA injection to evaluate the silencing efficiency.

**In vitro functional verification of GST_119 in Sf9 cell.** Sf9 cells were routinely maintained at 28 °C in SFX-insect cell culture medium supplemented with 10 % FBS and antibiotics (100 U/ml penicillin and 100 μg/ml streptomycin). The full length of GST_119 mRNA was amplified and cloned into pie-EGFP-N1 plasmid. Sf9 cells were seeded into 48-well plates and incubated for 24 h. The next day transfection was carried out using pie-EGFP-N1-GST_119 (0.5 μg/well) and Cellfectin (1 μl/well), blank pie-EGFP-N1 plasmid as a control. At 24 h post tranfection, the medium in the wells was removed and the cells were washed twice with PBS. The lambda-cyhalothrin (10 μg/ml, which have confirmed cytotoxicity to Sf9 cells[65]) was added in cultures for 48 h. The growth characteristics of Sf9 were observed under microscope and the number of cells (three different view).

**Interaction of GST and *lambda*-cyhalothrin using molecular docking.** The molecule structure of *lambda*-cyhalothrin (Compound CID: 6440557) was downloaded from PubChem database. Molecular docking of the GST_119 with *lambda*-cyhalothrin was conducted by the program SwissDock[66]. The binding mode structures of GST_119 with *lambda*-cyhalothrin were visualized using PyMOL.

**Cloning of GSTs promoter sequences.** Genomic DNA was extracted from larvae using a DNA isolation reagent (TaKaRa, China). The promoters were cloned using the Genome Walker Universal Kit (Clontech, USA) according to the manufacturer's instructions. Two-step PCR was performed, and then, the PCR products were cloned into the pMD-19T vector for sequencing. Transcription factor (TF) binding sites were predicted and analyzed using the online tools JASPAR and ALGGEN[67,68].

**Luciferase reporter assays.** The transcriptional activity of the promoter sequence of the GST genes was detected using the luciferase reporter assay in transiently transfected Sf9 cells. DNA fragments in front of the start codon of the translation of GST genes were cloned using genomic DNA as a template, and specific primer pairs were designed for cloning. After the start codon of the translation of ATG was mutated to CTG, it was inserted into the luciferase reporter plasmid pGL3-Basic. Internal plasmid pIE2-RL was used for the normalization of luciferase activity[69]. Sf9 cells were cultured in 48-well plates. After 12 h of culture, 255 ng mixed plasmids (pGL3-Basic-GST promoter + pIE2-RL) were used for the transfection of each well using the transfection reagent FuGENE HD (Promega, USA). The promoter activities were assayed using a dual-luciferase reporter kit (Promega, USA) according to the manufacturer's instructions. The plasmid pIE2-HaGATAe-His was constructed previously[69]. The co-transfection of plasmids pGL3-Basic-GST promoter, pIE2-HaGATAe-His, and pIE2-RL was also carried out. The empty vector pIE2-His was used as a control.

**Statistics and reproducibility.** Data analysis was performed using SPSS 16.0 software (SPSS, USA). The differences between the two samples were analyzed using the Student's *t* test. A significant difference was considered at $P < 0.05$. Values are represented as the mean ± standard error (SE) of at least three independent biological replicates.

**Reporting summary**. Further information on research design is available in the Nature Portfolio Reporting Summary linked to this article.

## Data availability

The whole genome resequencing data used in this paper are publicly available data under accession number SAMN18253696 and PRJNA713413[33]. The source data behind the graphs in Figs. 3, 5 and 6 can be found in Supplementary Data 1. All other data are available from the corresponding author on reasonable request. An uncropped gel image of Fig. 5 can be found in Supplementary Fig. 9. Promoter sequences of GST_119 and GST_121 in XJ and NTC population can be found in Supplementary Table 2.

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

## Acknowledgements

This project was funded by the Sci-Tech Innovation 2030 Agenda (2022ZD04021), National Natural Science Foundation of China (32372546), Shenzhen Science and Technology Program (KQTD20180411143628272), the Agricultural Science and Technology Innovation Program of Chinese Academy of Agricultural Sciences and Major projects of basic research of Science, Shenzhen Science and Technology Project (JCYJ20190813115612564), and Technology and Innovation Commission of Shenzhen Municipality.

## Author contributions

Y.T.X., M.H.J. conceived and designed the study; M.H.J. and Y.P. analyzed the data; H.H.Z and K.Y.L. performed the promoter experiments; M.H.J, J.P. and Y.X.S performed the insect experiments. M.H.J. and Y.T.X. wrote the manuscript along with K.Y.L. with input from all authors.

## Competing interests

The authors declare no competing interests.
