## [Peer Review File · Communications Biology]

Reviewers' comments:

Reviewer #1 (Remarks to the Author):

This manuscript explores the functional role of a GST cluster in the cotton bollworm, *Helicoverpa armigera*. In this manuscript, Jin et al. employed population genomic analysis to identify a GST cluster that was positively selected in a natural population exhibiting insecticide resistance. Additionally, genetic linkage analysis demonstrated the GST cluster was genetically linked to lambda-cyhalothrin resistance. The function of this cluster was validated through CRISPR/Cas9 knockout of a 26-kb chromosome region, which resulted in the loss of resistance. Moreover, gene expression and promoter analysis revealed that promoter variation mediated the up-regulation of GST in the North China population, consequently leading to insecticide resistance.

Based on the population genomic data, this article provides a comprehensive description of the role of the GST gene cluster in conferring insecticide resistance in the natural population of cotton bollworm. This study examines the evolutionary patterns, gene cluster expansion, and adaptive evolution associated with this GST cluster. This article presents a fascinating exploration of the topic, making it highly suitable for publication in *Communications Biology*.

I have some minor corrections and questions, which the authors could address in a minor revision.

Specific comments:

In Figure 2A, there are still five genes in the 2.71-3.01 Mb. How to exclude these genes not related to insecticide resistance? Are these genes also subject to selection pressure? In Figure 2C, the tightly linked GST cluster in the NTC population is clearly visible. It would be beneficial to provide the detailed LD R² value.

In the section on genetic linkage, it is necessary to provide additional detail on the construction of the F₂ population. The concentrations should be listed in line 150.

Line 168, the author obtained a 26 kb deletion homozygous knocked out the GST cluster. The selection process should be added to the methods. What's the knockout efficiency of chromosome deletion?

In this study, CncC/Maf transcription factor binding site was found in the promoter of GST₁₁₉ from the NTC population. Thus the authors discussed the CncC/Maf-mediated transcription regulation of the expression of GST genes. Apart from CncC/Maf, AhR/ARNT regulate the expression of P450 and GST have been documented recently. The following reference might be helpful: J. Agric. Food Chem. 2023, 71, 5230–5239; Pestic. Biochem. Physiol. 2021, 176,104875;

Other comments:

1. Line 27, “related” change to “genetic linked”
2. Line 71, “world’s major agricultural pests” change to “major agricultural pest worldwide”.
3. Line 76, “The insecticide resistance evolution of cotton bollworm” change to “The evolution of insecticide resistance in cotton bollworm”.
4. Line 104, this sentence should be deleted in the results section. The author can discuss this phenomenon in the Discussion.
5. Line 112, explain $\theta\pi$
6. Line 117, “confirming” change to “confirmed”
7. Line 175-178, this sentence is too complex and can easily mislead information.
8. Line 195, “the injection of two equally mixed dsRNAs did not show a synergistic effect”, the detoxification enzymes can perform functions independently, its normal that no synergistic effect between two GST genes. However, this is a good point to validate.
9. Line 334, cite a reference of insect strain.
10. Line 431, add a space between numerical values and units, check the whole manuscript.

Reviewer #2 (Remarks to the Author):

Recommendation: Major review.

The present paper used several experiments to prove the role of GST-119 in *H.armigera* against lambda-cyhalothrin (non-systemic pyrethroid insecticide) resistance. The paper can potentially be published in the current journal after resolving a few shortcomings. Overall, writing is ok but can be better and more engaging.

Major:

1. Introduction: Insecticide resistance is a severe problem that significantly challenges pest control. One of the most effective resistance mechanisms is the overexpression of genes coding for detoxification enzymes. A series of transcription factors control the expression of detoxification genes. The introduction did not mention the possibilities of mutation or changes in the regulatory regions (i.e., promoter) of detoxifying genes, transcription factors, or epigenetic modulators underlying insecticide resistance. I have given some example studies to consider, refer and enrich the introduction.

Example studies:

<https://pubmed.ncbi.nlm.nih.gov/28854876/>

<https://www.sciencedirect.com/science/article/abs/pii/S0965174818301681?via=ihub>

doi- 10.1016/j.j.cris.2021.100018

<https://parasitesandvectors.biomedcentral.com/articles/10.1186/s13071-020-04383-w>

<https://journals.plos.org/plosgenetics/article?id=10.1371/journal.pgen.1009403>

<https://journals.plos.org/plosgenetics/article?id=10.1371/journal.pgen.1010037>

<https://www.frontiersin.org/articles/10.3389/fphys.2019.01068/full>

2. The introduction also needs to introduce lambda-cyhalothrin (non-systemic pyrethroid insecticide) to the readers and its mode of action as an insecticide. Also, the resistance mechanism of lambda-cyhalothrin in other insects sets the background for the current research and experimental plans.

3. What is the hypothesis of the present study? It needs to be added in the introduction section.

4. I understood that the authors looked at the GST cluster based on an association study. What about the other interesting gene families capable of conferring resistance to insecticides? In the discussion, the authors acknowledge the contribution of Cytochrome p450 genes in cotton bollworms for their ability to metabolize xenobiotic substances. It will be better to add the gene expression profile of other detoxifying genes from the RNA-seq study in the supple, citing the 2019 paper on RNA-seq of *H. armigera*. Use FDR-corrected p-value for significant DGEs.

5. I do miss a small RNA-seq study on NTC and XJ strains. Understanding the overall gene expression dynamics under different resistance scenarios might be so much more informative. I am wondering why the authors did not do that.

6. It is worth overexpressing the GST-119 in the SF-9 cell line and checking for insecticide resistance in the SF9 cells. This will ultimately prove the contribution of overexpression of GST-119 underlying lambda-cyhalothrin resistance. I assume authors can do this easily.

7. Authors should perform the molecular docking study with candidate GSTs and the lambda-cyhalothrin (non-systemic pyrethroid insecticide) to evaluate the binding kinetics to support their claim about GST-119.

8. I do suggest to modify the discussion for better coherence. There is lack of appropriate citations at places.

9. Methodology: "The *H. armigera* laboratory strain 96S was started from 20 pairs of adults, collected from a conventional cotton field located in Xinxiang, Henan Province, China, in 1996." – where did the authors use this lab strain in the present paper? Do I miss something here. How do the authors

compare the lab strain reared <25 years in controlled environment with the field strain collected in 2021.

10. Title: Add (Lepidoptera: Noctuidae) after *H.armigera*. Consider restructuring the title. Try to add “Transcriptional regulation and overexpression of GST-cluster” instead of “Comprehensive regulation of GST gene cluster”.

Minor:

1. Line 29: KO strain?
2. Line 35 – what is XJ_GST_119? Write XJ in full or inside (). It is necessary to explain the short forms at its first appearance in the manuscript.
3. Remove “the” before North China.
4. Line 56: replace with – “Mutation of these targets often leads to higher resistance in pests”
5. Figure 2: Blood? I think the authors mean “haemolymph”. “Relative gene expression levels of GST genes in different developmental stages and tissues are shown using FPKM values” – it is from *H.armigera*. So, mention it in the legend to increase clarity.
6. Figure 3D: level “p-values” in the x-axis.
7. Line 143: delete “The”
8. Line 186: Other genes were low/not expressed or showed no significant change in expression. – what do you mean by other genes? Be precise.
9. Supplementary Figure S5, S6 resolution is poor and not readable at all.
10. Line 305- “it participates in the generation of insecticide resistance”- citation missing. Consider citing some of the previously suggested studies from comment one as suitability.
11. 2- $\Delta\Delta$ Ct method – citation missing.
12. Actin and GAPDH genes as internal controls – citation missing.

Reviewer #3 (Remarks to the Author):

The present study elucidates the nuances of GST gene cluster expression under two different selection pressure regimes and employed RNAi and CRISPR/Cas9 to elucidate the functionality of these genes in *H. armigera*. Also determined the prevalence of mutations in differential contribution of resistance to insecticides.

In this regard, whether or not in vitro cleaving/restriction assay was carried out with sgRNA1 and sgRNA2 before taking up in vivo studies? If so, please provide the gel picture supporting the large restriction observed in vivo.

Since the NHEJ yields random mutations, what were the other types of edits observed in G0?

Why sgRNAs were not tested for the promoter region?

Point-by-Point Response to Comments from the Reviewers

Reviewer #1 (Remarks to the Author):

This manuscript explores the functional role of a GST cluster in the cotton bollworm, *Helicoverpa armigera*. In this manuscript, Jin et al. employed population genomic analysis to identify a GST cluster that was positively selected in a natural population exhibiting insecticide resistance. Additionally, genetic linkage analysis demonstrated the GST cluster was genetically linked to lambda-cyhalothrin resistance. The function of this cluster was validated through CRISPR/Cas9 knockout of a 26-kb chromosome region, which resulted in the loss of resistance. Moreover, gene expression and promoter analysis revealed that promoter variation mediated the up-regulation of GST in the North China population, consequently leading to insecticide resistance.

Based on the population genomic data, this article provides a comprehensive description of the role of the GST gene cluster in conferring insecticide resistance in the natural population of cotton bollworm. This study examines the evolutionary patterns, gene cluster expansion, and adaptive evolution associated with this GST cluster. This article presents a fascinating exploration of the topic, making it highly suitable for publication in *Communications Biology*.

REPLY: We greatly appreciate the positive feedback.

I have some minor corrections and questions, which the authors could address in a minor revision.

Specific comments:

In Figure 2A, there are still have five genes in the 2.71-3.01 Mb. How to exclude these genes not related to insecticide resistance? Are these genes also subject to selection pressure? In Figure 2C, the tightly linked GST cluster in the NTC population is clearly visible. It would be beneficial to provide the detailed LD R² value.

REPLY: Thank you for the comments. During the selection pressure analysis, we calculated the $\theta\pi$, θ_w and Tajma's D in 10 Kb non-overlapping windows in the 300 Kb region. In the analysis of Tajma's D, the results showed that in addition to the GST cluster, three other genes were included in the selection region. In the analysis of $\theta\pi$, θ_w , only the GST cluster were showed significant selection pressure. Combined with these results, we excluded other genes that did not show selection pressure in the populations. The LD R² value of North China is 0.62, while that of Xinjiang is 0.15. This indicated that the GST cluster was more tightly linked in the North China population than in the Xinjiang population. The LD R² values have been added to the main text (line 149).

In the section on genetic linkage, it is necessary to provide additional detail on the constriction of the F₂ population. The concentrations should be listed in line 150.

REPLY: Revised as suggested (line 416-418). The diagnostic concentration of insecticides PX, LC and EB were added in the Methods (line 421-422)

Line 168, the author obtained a 26 kb deletion homozygous knocked out the GST cluster. The selection process should be added to the methods. What's the knockout efficiency of chromosome deletion?

REPLY: Revised as suggested (line 467-473). We knew that the efficiency of deleting chromosome fragments was not very high, so we injected a large number of eggs. Different knockout types were detected, mainly 6 types (we have added these mutation types in the Supplementary file, Figure S2). We estimated the knockout efficiency to be less than 5% (only the chromosome fragment deletions were counted).

Figure S2. Differently mutation types of GST cluster knockout. Target sequences of sgRNAs were marked with black and red lines.

In this study, CncC/Maf transcription factor binding site was found in the promoter of GST₁₁₉ from the NTC population. Thus the authors discussed the CncC/Maf-mediated transcription regulation of the expression of GST genes. Apart from CncC/Maf, AhR/ARNT regulate the expression of P450 and GST have been documented recently. The following reference might be helpful: J. Agric. Food Chem. 2023, 71, 5230–5239; Pestic. Biochem. Physiol. 2021, 176, 104875;

REPLY: Thank you for the comments. In the present study, we found that mutations in the promoter regions of GST could lead to the differential expression between populations. We only performed the validation experiments using the entire promoter regions. The prediction of TF binding sites in the promoter regions did not identify the binding site of AhR/ARNT. It needs more evidence to clarify the true TF binding sites and more experiments to verify its function in the future. We have discussed TF AhR/ARNT in the Discussion.

Other comments:

1. Line 27, “related” change to “genetic linked”

REPLY: Revised (line 27).

2. Line 71, “world’s major agricultural pests” change to “major agricultural pest worldwide”.

REPLY: Revised (line 80-81).

3. Line 76, “The insecticide resistance evolution of cotton bollworm” change to “The evolution of insecticide resistance in cotton bollworm”.

REPLY: Revised (line 84-85).

4. Line 104, this sentence should be deleted in the results section. The author can discuss this phenomenon in the Discussion.

REPLY: Revised.

5. Line 112, explain $\theta\pi$

REPLY: Revised (line 130).

6. Line 117, “confirming” change to “confirmed”

REPLY: Revised (line 135).

7. Line 175-178, this sentence is too complex and can easily mislead information.

REPLY: Combining the comments of other reviewers, we moved this sentence to the Discussion and refined it.

8. Line 195, “the injection of two equally mixed dsRNAs did not show a synergistic effect”, the detoxification enzymes can perform functions independently, its normal that no synergistic effect between two GST genes. However, this is a good point to validate.

REPLY: Thanks for your approbation.

9. Line 334, cite a reference of insect strain.

REPLY: Revised as suggestion.

10. Line 431, add a space between numerical values and units, check the whole manuscript.

REPLY: Revised as suggestion.

Recommendation: Major review.

The present paper used several experiments to prove the role of GST-119 in *H. armigera* against lambda-cyhalothrin (non-systemic pyrethroid insecticide) resistance. The paper can potentially be published in the current journal after resolving a few shortcomings. Overall, writing is ok but can be better and more engaging.

REPLY: We greatly appreciate the positive feedback.

Major:

1. Introduction: Insecticide resistance is a severe problem that significantly challenges pest control. One of the most effective resistance mechanisms is the overexpression of genes coding for detoxification enzymes. A series of transcription factors control the expression of detoxification genes. The introduction did not mention the possibilities of mutation or changes in the regulatory regions (i.e., promoter) of detoxifying genes, transcription factors, or epigenetic modulators underlying insecticide resistance. I have given some example studies to consider, refer and enrich the introduction.

Example studies:

<https://pubmed.ncbi.nlm.nih.gov/28854876/>

<https://www.sciencedirect.com/science/article/abs/pii/S0965174818301681?via=ihub>

doi- 10.1016/j.cris.2021.100018

<https://parasitesandvectors.biomedcentral.com/articles/10.1186/s13071-020-04383-w>

<https://journals.plos.org/plosgenetics/article?id=10.1371/journal.pgen.1009403>

<https://journals.plos.org/plosgenetics/article?id=10.1371/journal.pgen.1010037>

<https://www.frontiersin.org/articles/10.3389/fphys.2019.01068/full>

REPLY: Thanks for your comments. We have added a paragraph in the Introduction to describe the variants that may cause changes in gene expression (line 71-79).

2. The introduction also needs to introduce lambda-cyhalothrin (non-systemic pyrethroid insecticide) to the readers and its mode of action as an insecticide. Also, the resistance mechanism of lambda-cyhalothrin in other insects sets the background for the current research and experimental plans.

REPLY: Thanks for your suggestions. We have added these information in the Introduction (line 89-96).

3. What is the hypothesis of the present study? It needs to be added in the introduction section.

REPLY: We have added this information in the main text (line 96-100).

4. I understood that the authors looked at the GST cluster based on an association study. What about the other interesting gene families capable of conferring resistance to insecticides? In the discussion, the authors acknowledge the contribution of Cytochrome

p450 genes in cotton bollworms for their ability to metabolize xenobiotic substances. It will be better to add the gene expression profile of other detoxifying genes from the RNA-seq study in the supple, citing the 2019 paper on RNA-seq of *H. armigera*. Use FDR-corrected p-value for significant DGEs.

REPLY: We fully agree with you that genes such as CYP450 are also involved in insecticide resistance. However, the present study is primarily based on our previous research, where we identified a GST cluster from the perspective of forward genetics. We found that this GST cluster was under natural selection and carried out a series of verification experiments. For other genes of interest, such as CYP450, we will continue to analyse them and combine the information on population variation to address other interesting works.

5. I do miss a small RNA-seq study on NTC and XJ strains. Understanding the overall gene expression dynamics under different resistance scenarios might be so much more informative. I am wondering why the authors did not do that.

REPLY: Thank you for your comment. Small RNAs have significant roles in organisms, and reports indicate their involvement in insecticide metabolism/resistance. The comparison between NTC and XJ populations should reveal many interesting differentially expressed genes. However, this study mainly focuses mainly on the GST cluster, which is under strong selection in the NTC population. We performed genetic linkage and mutation verification to investigate its potential role in insecticide resistance. Various factors may contribute to resistance, including the small RNA or CYP450s you mentioned earlier, which are crucial for insecticide metabolism/resistance. We will take note of these directions in our future research. Thank you again for your valuable suggestions.

6. It is worth overexpressing the GST-119 in the SF-9 cell line and checking for insecticide resistance in the SF9 cells. This will ultimately prove the contribution of overexpression of GST-119 underlying lambda-cyhalothrin resistance. I assume authors can do this easily.

REPLY: Thanks for the suggestions. Several studies have shown that lambda-cyhalothrin is cytotoxic to Sf9 cells. The Sf-9 cell line is suitable for the toxicity validation experiments. The GST_119 plasmid was constructed and transfected into Sf9 cells. A blank plasmid was also transfected as a control. After incubation with lambda-cyhalothrin, we found that Sf9 cells transfected with GST_119 plasmid had a significantly higher survival rate than Sf9 cells transfected with blank plasmid ($p = 0.029$). We have added these results in Results (line 212-217), Methods (line 506-517) and Supplementary files (Figure S5).

Figure S5. Cytotoxicity of *lambda*-cyhalothrin to Sf9 cells after transfected GST_119 plasmid and blank plasmid.

7. Authors should perform the molecular docking study with candidate GSTs and the lambda-cyhalothrin (non-systemic pyrethroid insecticide) to evaluate the binding kinetics to support their claim about GST-119.

REPLY: Thanks for the suggestions. We have performed the molecular docking analysis. The results were added in the Results (line 217-220), Methods (line 518-522) and Supplementary files (Figure S6).

Figure S6. Interaction analysis between GST_119 proteins and *lambda*-cyhalothrin.

8. I do suggest to modify the discussion for better coherence. There is lack of appropriate citations at places.

REPLY: Thanks for the suggestions. We have revised the section of Discussion.

9. Methodology: “The *H. armigera* laboratory strain 96S was started from 20 pairs of adults, collected from a conventional cotton field located in Xinxiang, Henan Province, China, in 1996.” – where did the authors use this lab strain in the present paper? Do I miss something here. How do the authors compare the lab stain reared <25 years in controlled environment with the field strain collected in 2021.

REPLY: The lab strain 96S is a pesticide-sensitive strain used as a control for the North China field collection resistance strain and the GST knockout strain. Fold resistance is a relative value, whereas the 96S strain is used as a sensitive line to calculate relative fold resistance in the NTC and GST KO strains (line 194 and Figure 5C). To make this clear, we have added a statement in the Methods explaining that the strain used for the knockout is the NTC strain. Pesticide resistance mechanism in cotton bollworm are complex. After knocking out the GST gene cluster, we found that the knockout strain was still somewhat resistant compared to 96S. We are not sure whether this is due to the different genetic backgrounds of the strains or whether it suggests that there are other resistance mechanisms besides the GST cluster exist in the North China population. We discussed this phenomenon in the Discussion.

10. Title: Add (Lepidoptera: Noctuidae) after *H.armigera*. Consider restructuring the title. Try to add “Transcriptional regulation and overexpression of GST-cluster” instead of “Comprehensive regulation of GST gene cluster”.

REPLY: Thanks, revised as suggestion.

Minor:

1. Line 29: KO strain?

REPLY: Revised to knockout strain (line 29).

2. Line 35 – what is XJ_GST_119? Write XJ in full or inside (). It is necessary to explain the short forms at its first appearance in the manuscript.

REPLY: Revised (line 36).

3. Remove “the” before North China.

REPLY: Revised.

4. Line 56: replace with – “Mutation of these targets often leads to higher resistance in pests”

REPLY: Revised (line 56-57).

5. Figure 2: Blood? I think the authors mean “haemolymph”. “Relative gene expression levels of GST genes in different developmental stages and tissues are shown using FPKM values” – it is from *H.armigera*. So, mention it in the legend to increase clarity.

REPLY: We have revised according to your suggestion (Figure 3B, line 747)

6. Figure 3D: level “p-values” in the x-axis.

REPLY: Revised (Figure 4D).

7. Line 143: delete “The”

REPLY: Revised .

8. Line 186: Other genes were low/not expressed or showed no significant change in expression. – what do you mean by other genes? Be precise.

REPLY: Other genes including GST_116, GST_118 and GST_120. We have added it in the main text (line 204).

9. Supplementary Figure S5, S6 resolution is poor and not readable at all.

REPLY: Thanks for your reminding. We have replaced higher resolution figures (Figure S7 and S8).

10. Line 305- “it participates in the generation of insecticide resistance”- citation missing. Consider citing some of the previously suggested studies from comment one as suitability.

REPLY: Revised .

11. 2- $\Delta\Delta$ Ct method – citation missing.

REPLY: Revised .

12. Actin and GAPDH genes as internal controls – citation missing.

REPLY: Revised .

Reviewer #3 (Remarks to the Author):

The present study elucidates the nuances of GST gene cluster expression under two different selection pressure regimes and employed RNAi and CRISPR/Cas9 to elucidate the functionality of these genes in *H. armigera*. Also determined the prevalence of mutations in differential contribution of resistance to insecticides.

In this regard, whether or not in vitro cleaving/restriction assay was carried out with sgRNA1 and sgRNA2 before taking up in vivo studies? If so, please provide the gel picture supporting the large restriction observed in vivo.

REPLY: Thanks for your comments. We have not performed an in vitro cleavage assay. Verification of sgRNA cleavage activity in vitro and selection of sgRNA combinations with high cleavage activity is likely to increase editing efficiency. The research subject in this study is the cotton bollworm, which lays a substantial number of eggs of a moderate size, making it more amenable to microinjection. We have established a reliable system for chromosome fragment knockout experiments (DOI 10.1111/1744-7917.12570; DOI 10.1002/ps.6170). Therefore, we skipped the in vitro validation process and proceeded directly to the in vivo experiment.

Since the NHEJ yields random mutations, what were the other types of edits observed in G0?

REPLY: Thanks for the comments. In total, six different mutation types were detected. We have added the statement in the main text (line 187) and added a supplemental figure (Figure S2).

Figure S2. Differently mutation types of GST cluster knockout. Target sequences of sgRNAs were marked with black and red lines.

Why sgRNAs were not tested for the promoter region?

REPLY: Thanks for the comments. Frankly, we did not attempted to knock out the promoter region. In the present study, we mainly focused on the natural selection-mediated variation in populations that causes differences in pesticide susceptibility in different populations. By amplifying the promoter regions of the respective populations, we found that the promoter sequences varied significantly between NTC and XJ. We then selected a traditional method for detecting promoter activity - the dual luciferase activity assay. The results showed a significant dissimilarity between the activities, indicating that the difference in promoter activity triggered variation in GST gene expression. Thank you again for your comments and we will consider knocking out the promoter region in future studies.

REVIEWERS' COMMENTS:

Reviewer #1 (Remarks to the Author):

The authors have addressed all of my concerns. In my opinion, the manuscript can be accepted for publication.

Reviewer #2 (Remarks to the Author):

The authors successfully answered all my queries and performed the necessary experiments, which indicated that over-expression of GST-119 is a crucial player in lambda-cyhalothrin resistance in *H. armigera*. I have no more queries and i think this manuscript now ready to be published.

I want to congratulate the authors for this nice work and request them to follow up on some of the findings in future to better understand the pesticide resistance mechanisms in pest insects.

Reviewer #3 (Remarks to the Author):

The authors have clarified all the queries and the MS can be published in the present form in Communications Biology